# Investigation of the Interaction between Mechanosynthesized ZnS Nanoparticles and Albumin Using Fluorescence Spectroscopy

**DOI:** 10.3390/ph16091219

**Published:** 2023-08-29

**Authors:** Zdenka Lukáčová Bujňáková, Erika Dutková, Jana Jakubíková, Danka Cholujová, Rastislav Varhač, Larysa Borysenko, Inna Melnyk

**Affiliations:** 1Institute of Geotechnics, Slovak Academy of Sciences, Watsonova 45, 04001 Košice, Slovakia; dutkova@saske.sk (E.D.); melnyk@saske.sk (I.M.); 2Cancer Research Institute of Biomedical Research Center, Slovak Academy of Sciences, Dúbravská Cesta 9, 84505 Bratislava, Slovakia; jana.jakubikova@savba.sk (J.J.);; 3Faculty of Science, Pavol Jozef Šafárik University in Košice, Šrobárova 2, 04154 Košice, Slovakia; rastislav.varhac@upjs.sk; 4Chuiko Institute of Surface Chemistry, National Academy of Science of Ukraine, Generala Naumova 17, 03164 Kyiv, Ukraine; borysenko@yahoo.com

**Keywords:** ZnS nanoparticles, albumin, milling, fluorescence, quenching

## Abstract

In this paper, ZnS nanoparticles were bioconjugated with bovine serum albumin and prepared in a form of nanosuspension using a wet circulation grinding. The stable nanosuspension with monomodal particle size distribution (d_50_ = 137 nm) and negative zeta potential (−18.3 mV) was obtained. The sorption kinetics and isotherm were determined. Interactions between ZnS and albumin were studied using the fluorescence techniques. The quenching mechanism, describing both static and dynamic interactions, was investigated. Various parameters were calculated, including the quenching rate constant, binding constant, stoichiometry of the binding process, and accessibility of fluorophore to the quencher. It has been found that tryptophan, in comparison to tyrosine, can be closer to the binding site established by analyzing the synchronous fluorescence spectra. The cellular mechanism in multiple myeloma cells treated with nanosuspension was evaluated by fluorescence assays for quantification of apoptosis, assessment of mitochondrial membrane potential and evaluation of cell cycle changes. The preliminary results confirm that the nontoxic nature of ZnS nanoparticles is potentially applicable in drug delivery systems. Additionally, slight changes in the secondary structure of albumin, accompanied by a decrease in α-helix content, were investigated using the FTIR method after analyzing the deconvoluted Amide I band spectra of ZnS nanoparticles conjugated with albumin. Thermogravimetric analysis and long-term stability studies were also performed to obtain a complete picture about the studied system.

## 1. Introduction

Zinc sulfide (ZnS) is an important type of semiconductor belonging to II-VI group. ZnS is a mineral compound that exists in two phases, hexagonal wurtzite and cubic sphalerite. Because of its wide optical band gap (3.68 eV) it is considered to be promising for applications such as light emitting diodes [1], sensors, lasers [2], photovoltaic or solar cells [3]. In the form of nanoparticles, ZnS also offers opportunities in biomedical applications, especially for targeted drug delivery systems or as an imaging agent [4]. Different methods have been used to prepare ZnS nanoparticles such as microwave [5], hydrothermal [6], solvothermal [7] and mechanochemical methods [8,9]. Mechanochemistry is a branch of chemistry which is concerned with the chemical and the physicochemical changes that occur in substances of all states of aggregation due to the influence of mechanical energy. Mechanochemistry belongs to the green technologies and fully meets the criteria of the twelve principles of green chemistry [10]. Therefore, in recent years, mechanochemistry has been growing into a widely accepted alternative for chemical synthesis. The “rediscovery” of mechanochemistry has occurred as it represents an alternative method for cleaner, safer, more efficient and sustainable transformations in the chemical and pharmaceutical industries.

However, bare ZnS as an inorganic compound is a less appropriate material for bioconjugation, and subsequently for drug delivery and bioimaging, because its surface is less accessible for bioconjugate reactions and is unstable against chemical reactions. Thus, surface modification with biocompatible molecules is required. Coating or conjugation of such molecules on a ZnS surface has become an inevitable step for extended stability and biocompatibility and for protection of nanoparticles against hydrolysis and biochemical reactions [11]. Generally, there are two types of functional coatings, organic and inorganic. Inorganic coatings are used to link biological compounds at the surface of nanoparticles and enhance their antioxidant properties. Silica, carbon, metals, metal oxides and sulfides can be mentioned as examples [12]. Organic coatings increase dispersibility and biocompatibility and have been used for specific drug targeting. They can be classified into three groups: (i) small molecules and surfactants, (ii) macromolecules and polymers and (iii) biological molecules. For instance, ZnS nanoparticles were coated with nonionic surfactants such as sorbitan monolaureate and sorbitan monooleate to prepare stable emulsions [13]. A highly stable nanosuspension of ZnS coated with chitosan, exhibiting interesting fluorescence properties was synthesized by the mechanochemical approach [14]. Among the biological molecules, bovine serum albumin (BSA) is an extensively used as biocompatible template for the synthesis of nanoparticles due to the presence of abundant functional groups in BSA, such as hydroxyl, amine, carboxyl and thiol. Moreover, albumin is the most abundant protein in blood plasma and plays a dominant role in controlling the distribution, excretion, therapeutic efficacy and toxicity of numerous endogenous and exogenous ligands in the body [15]. ZnS nanoparticles and nanocomposites coated with albumin were prepared by many researchers [16,17,18,19]. The interaction between these entities was also studied [20,21] with the evaluation of toxicity assessment [22].

The potential of ZnS nanoparticles in a drug delivery system has been investigated through their bioconjugation. For instance, Abniki et al. [23] performed adsorption studies of famotidine by 3-aminophenol-grafted AGE/ZnS. Famotidine is a poorly water-soluble drug and acts on the parietal cells to inhibit histamine H2-receptors. The successful novel ZnS-based nanocarrier for sustained release of famotidine was prepared by the authors. Li et al. [24] developed the CdSe/ZnS@PEI delivery system to carry the anticancer drug 10-hydroxycamptothecin. The formulation displayed an efficient therapeutic effect in inhibiting tumor growth in mice and demonstrated good biosafety and effectively enriched at the tumor site to improve the treatment effect, suggesting its significant potential for biomedical applications. Mn-doped ZnS nanoparticles were coated with cell penetrating peptide to load another anticancer drug, paclitaxel, as was reported by Rejinold et al. [25]. The obtained system showed an enhanced anticancer effect compared to bare paclitaxel, which has been validated by MTT assay followed by apoptosis assay and DNA fragmentation analysis. In vivo biodistribution and anticancer efficacy were also studied on a breast cancer xenograft model showing maximum tumor localization and enhanced therapeutic efficacy. According to the authors, it could be an appropriate theranostic nano-carrier for paclitaxel with enhanced therapeutic efficacy against cancer cells, where penetration and sustainability of therapeutics are essential. There are several more examples of ZnS nanoparticles research on drug delivery across the literature review.

One of the main research tools in biochemistry and biophysics is fluorescence spectroscopy [26]. It is possible to study biological molecules (e.g., proteins, enzymes), their interactions, activities and changes in cellular metabolisms using this method. Fluorescence techniques are widely used in diagnostics, imaging and fluorescent labeling. Fluorescence microscopy techniques allow the identification of cancer cells, assessment of organelle dysfunction and evaluation of cell response to various treatments, as well as transport pathways and modalities of how drugs enter cells. Moreover, fluorescence spectroscopy can be used for the study of the interactions between biological molecules and inorganic materials in the research and development of drug delivery systems. In recent years, the studies of such interactions are on the rise. For example, the interaction between silver and gold nanoparticles and BSA has been extensively investigated [27,28,29]. The reactions between Fe_3_O_4_ and BSA were also studied using the fluorescence method [30]. Similarly, ZnO nanoparticles and BSA were explored [31] along with the interactions between ZnS nanoparticles and BSA [20,21]. Many more examples of such studies exist. By analyzing the obtained fluorescence spectra, the quenching mechanisms, binding constant or stoichiometry of the binding process can be determined. Fluorescence spectroscopy was also used for the determination of toxicity of nanomaterials on biological molecules [22,32]. 

However, a limited number of studies have focused on the preparation technology for large-scale production of stable albumin-based nanoparticles with uniform particle size distribution [33,34,35,36,37]. The wet circulation grinding belongs to such methods and is widely used for the processing of poorly water-soluble drugs. In this paper, the highly stable nontoxic ZnS–albumin nanosuspension with the uniform particle size distribution was easily prepared using this approach. 

Here, ZnS nanoparticles were mechanochemically synthesized in the first step and, subsequently, the nanosuspension of ZnS bioconjugated with bovine serum albumin (ZnS–BSA) was prepared by a wet stirred media milling approach. Structural changes of albumin influenced by this interaction with ZnS nanoparticles were studied in depth using fluorescence spectroscopy. The quenching mechanism was determined, and parameters describing the interaction of BSA with mechanosynthesized ZnS nanoparticles were calculated (e.g., quenching rate constant, binding constant, stoichiometry of binding process, accessibility of fluorophore to quencher). The determination of the changes in the secondary and tertiary structure of albumin was also the aim of this study. Fourier transform infrared spectroscopy and circular dichroisms spectroscopy were selected for this purpose. Moreover, the possible binding mechanisms between ZnS particles and albumin were outlined. At the end of this paper, the cellular mechanism of nanosuspension treatment in multiple myeloma cells was evaluated by fluorescence assays for quantification of apoptosis, assessment of mitochondrial membrane potential and DNA content analysis of nuclei labeled for evaluation of cell cycle changes.

## 2. Results and Discussion

### 2.1. Solid-State Properties of Mechanochemically Synthesized ZnS—Minireview

Briefly and most concisely, the ZnS nanocrystals were prepared by the comilling of zinc acetate and sodium sulfide (according Equation (1)) as was described in our previous work [8]. The XRD analysis confirmed the presence of both cubic (sphalerite) and hexagonal (wurtzite) phases. The structure of ZnS nanocrystals with the crystallite size of 2–10 nm was clearly identified using the Williamson–Hall analysis, the Warren–Averbach method [38] or the Rietveld refinement [39,40]. These results were in good accordance with HRTEM analysis. A large portion of nanoparticles were defective. The nanocrystals tend to aggregate; however, their surface uniformity and homogeneity were well documented. The UV–Vis absorption spectrum showed a blue shift in comparison to the bulk ZnS as a consequence of quantum confinement effect [38]. The nanoparticles possess a high value of specific surface area determined as 126 m^2^ g^−1^ [38,39,40]. More detailed structure and surface as well as optical properties information regarding to mechanochemically synthesized ZnS nanoparticles have been deeply described in papers [8,38,39,40] and the corresponding figures of XRD, HRTEM and UV-Vis absorption spectra are attached as Appendix A).

### 2.2. Sorption Kinetics and Isotherm

In the first step, we examined the time required for sorption equilibrium to be established in the aqueous system containing the ZnS sample and BSA. According to the findings in Figure 1a, the ZnS sample reached equilibrium in just 120 min; however, 82% of the total adsorbed is absorbed in 60 min. The kinetic curve was plotted against Equations (2) and (3) for the processes of pseudo-first and pseudo-second order, respectively. Correlation coefficient values (Table 1) showed the straightening patterns approximately for two equations. This behavior can be attributed to two parallel reactions occurring on their surfaces, with significantly different rates. As the sample possesses both Zn^2+^ and S^2−^ sites on its surface, the kinetic curve for it was described by both equations, suggesting the binding of BSA to various adsorption centers. The linearized plots for modeling adsorption kinetics are attached as Appendix A). No changes or shifts in the UV–Vis spectra are observed over time (Figure 1b), indicating consistent interactions between the adsorbent and adsorbate, irrespective of time.

In the second step, we evaluated the sorption capacity of the ZnS sample to BSA from the adsorption isotherm (Figure 2a and Table 1). We also plotted the sorption isotherm of BSA using the Langmuir (Equation (4)) and Freundlich (Equation (5)) equations, and the parameters of BSA sorption are presented in Table 1. As shown in Table 1, the experimental isotherm of the ZnS sample fits well with the Freundlich adsorption equation (Figure 2b), indicating that the surface of the adsorbent is heterogeneous and contains several types of available sites that act simultaneously. Therefore, we can infer that the adsorption processes on the ZnS sample occur through the formation of a monolayer based on the interaction with thiol groups of BSA and ZnS, along with the simultaneous involvement of other groups present on the ZnS surface and albumin.

### 2.3. Preparation of ZnS–BSA Nanosuspension

The ZnS–BSA nanosuspension was prepared within 45 min of milling. The particle size distribution measured by cross-correlation spectroscopy is shown in Figure 3a. Unimodal distribution was observed with the sizes of particles from 75 to 200 nm, and the average size of 137 nm.

Zeta potential (ZP) is one of the most important parameters for the stability determination of a colloidal system. The pH dependence of ZP for the pure ZnS (measured in distilled water, black line) and ZnS–BSA (red line) is shown in Figure 3b. In the case of ZnS particles dispersed in distilled water, it can be seen that the sample reached positive values of ZP in almost all the studied pH range. The highest value of ZP (+19 mV) was detected at pH 3. With increasing pH, the ZP reached fewer positive values and the isoelectric point (IEP) was determined at pH 7.3. Notably, this value is considerably higher in comparison with those described in other scientific papers reporting the IEPs below 3.0 [41], or at 3.0 [42], or the values in the range 3.0–3.5 were also obtained [43]. As we reported earlier [8,40], XRD results indicated both the phases, sphalerite and wurtzite in the mechanosynthesized ZnS. Liu at al. [41] described that the different crystal structure of ZnS causes the difference in their electrokinetic behaviors. Synthetic ZnS, which was identified as wurtzite by XRD technique, had an IEP of 8.5, while sphalerite had an IEP of less than 3.0. In summary, the positive zeta potential values are attributed to the Zn(II) ions present at the surface of the crystals (predominantly from the wurtzite structure) and their subsequent transfer into the water. In addition, in our case, a higher value of IEP is connected with the high specific surface area (126 m^2^ g^−1^) of the sample and, subsequently, a higher amount of active sites available for the dissolution of Zn(II) ions from the surface. Moreover, during the mechanochemical synthesis, a lot of defects are created, which are responsible for the increasing reactivity [44,45].

In the case of the ZnS–BSA sample, all the values of ZP shifted to more negative areas and the IEP was detected at 4.3. A very similar result was also obtained in a paper [20]; specifically, an IEP of 4.5 was reported after the adsorption of BSA onto ZnS nanoparticles prepared by the precipitation. Moreover, at working pH 7.4 the ZP of −18.3 mV was determined in comparison with −0.15 mV for uncoated ZnS. In general, albumin shows a negative ZP at a pH of human blood as a consequence of free carboxyl groups [46] and after its adsorption on the surface of the nanoparticles shifts ZP to a more negative value [47] as it was also detected in our case. As was summarized by Honary and Zahir [48], the negatively charged nanoparticles have some advantages in comparison to positive ones. The first one is slower clearance from the blood stream and the second one is lower cytotoxicity. In the light of these findings, the ZnS–BSA nanoparticles with a ZP of −18.3 mV are expected to be stable, making them a promising choice for potential applications.

### 2.4. Interaction Study between ZnS and BSA

#### 2.4.1. UV–Vis Spectroscopy

UV–Vis spectroscopy is an important tool for the studying of the molecular interactions between proteins (donor) and nanoparticles (acceptor). The absorption spectra of BSA with different ZnS concentrations are shown in Figure 4. Characteristic absorption peak of BSA is occurred at 278 nm. This peak is attributed to n-π* transition of the amino acid residues, such as tryptophan, tyrosine or phenylalanine [49]. As can be seen, the absorption intensity of the ZnS–BSA samples increases with raising the ZnS nanoparticle concentrations without any shifts in its absorption maximum, i.e., hyperchromism. In such a case, the microenvironment around the amino acid residues in BSA is not changed upon interaction with ZnS. The increase in the absorbance of BSA is due to the formation of a ground state complex between BSA and nanoparticles [19,50,51]. The absorption data have been analyzed using the Benesi–Hildebrand equation (see Equation (6)). A graph of 1/(A_obs_ − A_0_) versus 1/[c] yielded a linear plot with a slope equal to 1/K_app_(A_c_ − A_0_) and an intercept equal to 1/(A_c_ − A_0_). K_app_ was calculated to be 2 × 10^3^ M^−1^ and shows the binding affinity of ZnS with BSA (Figure 4 inset).

#### 2.4.2. Fluorescence Quenching Spectra

The fluorescence quenching refers to any process that decreases the fluorescence intensity of a sample. The quenching of fluorescence emission originates either from static or dynamic interaction of the quencher with the fluorophore. In order to obtain an insight into the quenching mechanism, the fluorescence spectra of BSA and BSA with various concentrations of ZnS nanoparticles have been observed upon excitation at 280 nm (Figure 5a). The emission band detected at 345 nm for all the samples is referred to as tryptophan residues in BSA [52,53]. The fluorescence intensity was found to decrease gradually with the increase in the quencher concentration (in opposite way to absorbance intensity), i.e., hypochromism. Our results correlate with those published in papers dealing with spectroscopic studies of the interaction between BSA and ZnS nanoparticles [20,21,54]. The ZnS nanoparticles quench the chromophore residue of albumin. Moreover, no shift in the peak position was detected. This could indicate that the nanoparticles do not induce changes in the tryptophan residues’ microenvironment and nonfluorescence ground state complex is created between them [55,56,57,58].

As was mentioned above, dynamic (diffusive collisions) or static (formation of a nonfluorescent ground state complex) interactions exist between quencher and fluorophore. For these purposes, the classical Stern–Volmer equation was used for the calculation of the Stern–Volmer constant *K_sv_* (see Equation (7)). The plots of F_0_/F versus [c] are presented when analyzing quenching data. It is expected a linear dependence of F_0_/F on the concentration of quencher. In such a case, a plot of F_0_/F versus [c] yields an intercept of one and a slope equals to *K_SV_.* However, in our case, the Stern–Volmer plots were found to be nonlinear. In many cases the fluorophore can be quenched both by static and dynamic interactions with the same quencher, as was described earlier [59]. In such a case, an upward curvature, concave towards the y-axis is observed (Figure 5b). Therefore, the Stern–Volmer constant was calculated from the initial linear part of regression curve appertaining lower quencher concentrations (below 7.3 × 10^−5^ M, see inset of Figure 5b). It points to the dynamic mechanism of quenching. At higher concentrations, positive deviation from linearity suggests a more complex quenching process, probably an additional presence of a static component in the quenching mechanism [60,61]. From the linear part of a regression, F_0_/F versus [c] the *K_SV_* constant 2 × 10^4^ M^−1^ was calculated. This value is higher in comparison with that reffered in the paper [20], for instance, where *K_SV_* 7.79 × 10^3^ L mol^−1^ between ZnS and human serum albumin was reported. On the other hand, it is smaller than that mentioned in [21], where *K_SV_* = 1.87 × 10^6^ L mol^−1^ was calculated. However, a very similar result for *K_SV_* was also published in [54]. 

Subsequently, the bimolecular quenching rate constant *k_q_* was calculated using Equation (8) to be 4.3 × 10^11^ M^−1^s^−1^. The constant *k_q_*, reflects the efficiency of quenching or the accessibility of the fluorophores to the quencher. Diffusion controlled quenching typically results in values of *k_q_* near 1 × 10^10^ M^−1^s^−1^ [26]. Smaller *k_q_* values can result from steric shielding of the fluorophore or a low quenching efficiency. Larger *k_q_* values indicate some type of binding interaction. According to our results, both dynamic and static quenching (with the prevailing generation of nonfluorescence complex between them—ground state complex) exist between BSA and ZnS nanoparticles. The static quenching was concluded in papers [19,54], where *k_q_* ~ 10^13^ M^−1^ s^−1^ was estimated.

#### 2.4.3. Stoichiometry of the Binding Process

For the calculation of the number of ZnS molecules (*m*) that interact simultaneously with each site of BSA and the apparent binding constant (*K_m_*), Equation (9) was used. A plot of log[(F_0_ − F)/F] versus log[c] gives a straight line, whose slope equals to *m* and the intercept equals to *logK_m_* (Figure 5c). The slope of the plot gives the stoichiometry of the quencher/protein complex and was found to be 1.51. The binding constant *K_m_* was calculated to be 3.1 × 10^6^ M^−1^. The very similar results (*n* = 1.5 and *K_m_* = 7.9 × 10^6^ M^−1^) were also reported in [54]. 

#### 2.4.4. Accessibility of BSA to ZnS Nanoparticles

The accessibility of BSA to the ZnS nanoparticles was also evaluated using the modified Stern–Volmer equation (see Equation (10)). The linear plot of F_0_/(F_0_ − F) versus 1/[c] yields 1/*f_a_* as the intercept, and 1/(K_a_f_a_) as the slope. The *f_a_* value was calculated as 1.73 (R^2^ = 0.9898, Figure 5d), which means greater than 100%. This result suggests that the process is probably more complicated than the modified Stern–Volmer equation assumes, as the obtained data were better described using a first order exponential decay (R^2^ = 0.9962, Figure 5e). In such a case at least two classes of fluorescence with different efficiencies of quenching are assumed [62]. The higher *f_a_* values were also described, e.g., due to the formation of complex(es) with multiple ligands resulting in an oversimplification of the equation [63], or due to the fraction of fluorophore molecules, which may exist in an ‘intermediate stage of sequestration’ [64]. Authors in a paper [65] referred to the high values of *f_a_*, which were connected with the high values of *K_a_*, which were significantly enhanced at higher temperatures. Moreover, they assumed that the *f_a_* was highly sensitive to the secondary structure of RNA. We also assume that the high value of *f_a_* in our case could be partially related to the changes in the secondary structure of the BSA after mechanical milling. Moreover, during this process, the formation of complexes between ZnS nanoparticles and BSA occurs as indicated by the positive deviation from the linearity of Stern–Volmer plots described above. These *f_a_* > 100% values are in many cases interpreted as 100% of the tryptophan fluorescence residues in BSA can be quenched. The *K_a_* value was calculated as 8.3 × 10^3^ M^−1^, log*K_a_* = 3.92. The results describing the fluorescence quenching process are summarized in Table 2.

#### 2.4.5. Synchronous Fluorescence Spectra

According to Miller [66], synchronous fluorescence spectra indicate the changes in the microenvironment of both tyrosine and tryptophan residues of BSA, when wavelength difference (Δλ) between excitation and emission monochromators is established at 15 nm and 60 nm, respectively. Figure 6 shows an increasing concentration of ZnS nanoparticles leads to a significant decrease in the fluorescence intensity. We have calculated that the decrease in fluorescence intensity for tyrosine residues (Figure 6a) was about 62%, and for tryptophan (Figure 6b) it was about 79%. It suggests that the fluorescence from tryptophan is more intense and is more efficiently quenched in the presence of ZnS, implying that it can be closer to the binding site than tyrosine residue [51]. No shifts in emission wavelengths for both fluorophores were evidenced.

#### 2.4.6. CD Spectroscopy

The far-UV CD spectra were conducted to monitor the changes in the secondary structure of BSA after its interaction with ZnS nanoparticles. Figure 7 shows the CD spectrum of free BSA (black line) and spectrum of the ZnS–BSA sample (red line). A typical spectrum of BSA exhibits two characteristic negative bands at 210 and 222 nm originating from α-helical π → π* and n → π* amide transitions, respectively. As we can see, no changes in spectra were detected. The spectra seem to be almost identical, indicating the conformation of the BSA remains almost unchanged after the preparation of the ZnS–BSA nanosuspension by wet circulation grinding process. This conclusion is very important from the technological viewpoint, as the structure of albumin has not been destroyed during the milling and its functionality has been preserved.

#### 2.4.7. Fourier Transform Infrared Spectroscopy—FTIR

The FTIR spectrum of pure BSA, ZnS–BSA after milling (nanosuspension) and ZnS–BSA after adsorption experiments is depicted in Figure 8a. The most important peaks of BSA are assigned. The peak located at 3370 cm^−1^ is assigned to –NH stretching vibration, overlapping with the vibration of –OH hydroxyl group. The symmetric and antisymmetric stretching of =C-H, CH_2_ and –CH_3_ bands of the alkane chains in the protein are located in the region from 3060 to 2875 cm^−1^. The peak at 1656 cm^−1^ is attributed to the carbonyl C=O stretching of the amide I band, the most sensitive probe for detecting changes in the protein secondary structures. The amide II band located at 1542 cm^−1^ can be assigned to the N-H bending vibrations and the amide III band located at 1248 cm^−1^ is due to C-N and N-H in-plane stretching vibration [29,67,68,69,70,71]. 

Comparing the FTIR spectra of ZnS–BSA with that of pure BSA, the changes in the intensity and positions in the characteristic IR bands can be noticed. The characteristic peak of –NH and –OH vibration groups shifts to a higher wavenumber of about 81 cm^−1^ (from 3370 to 3426 cm^−1^). The intensity of C-H vibrations was considerably suppressed, but they did not disappear completely. In the region of amide I band (Figure 8b), the changes in the strength and position (from 1656 to 1645 cm^−1^) have been also occurred after the interaction of BSA with ZnS. The deconvolution of the peaks attributed to the amide I was used to study these changes in detail (Figure 8c–e). The amide I was deconvoluted into four absorption bands corresponding to antiparallel β-sheet, α-helix, random coil and β-sheet. In pure BSA (Figure 8c), the calculated content of appropriate structures was as followed: 22% for antiparallel β-sheet, 72% for α-helix, 3% for random coil and 3% for β-sheet. In the case of ZnS–BSA after milling (Figure 8d), the slight differences in contents were detected: 10%, 66%, 10% and 14%, respectively. More dramatic changes in secondary structures have occurred in a sample after the adsorption experiment (Figure 8d). The α-helix content was decreased only to 3%. On the other hand, the content of the random coil strongly increased up to 91%. The results are summarized in Table 3.

The conformational changes at tertiary level were followed by the hydrogen–solution exchange of the partially unfolded protein through time evolution of Amide II band according to [72,73]. These changes are also manifested by a reduction in the Amide II band intensity with the simultaneous increasing in the Amide II’ band. The Amide II band of BSA in H_2_O solution is centered at 1542 cm^−1^, and the Amide II’ is centered at 1450 cm^−1^. After the zoom of the appropriate wavenumbers, the ratio between intensities of Amide II’ and Amide II bands for each sample were calculated confirming an increase and decrease in Amide II’ and Amide II bands, respectively, after the interaction of BSA with ZnS particles. Subsequently, from the ratio of the areas of Amide II and Amide I bands, the percentage of amide protons that are exchanged during the deuteration were quantified. The results suggest that the BSA has partially unfolded, and hydrogens can undergo hydrogen–solution exchange up to 53 and 63% in samples after milling and after adsorption, respectively. 

In comparison to our previous results where the interaction between BSA and As_4_S_4_ nanoparticles was studied [37], the results in changes of secondary and tertiary structures are different. Dramatic changes in secondary structure were noticed for the ZnS–BSA sample after adsorption, where BSA has adsorbed on coarse, aggregated ZnS particles. The random coil almost completely substituted other structures, which means that the BSA has lost its helix structure at the interface and turned into disorder protein. In the case of the tertiary structure of As_4_S_4_-BSA sample after milling (fine particles) the hydrogen–solution exchange of BSA was determined only to be 19%. Now, for the ZnS–BSA sample, the deuteration was calculated to be 53%. On the other hand, after the adsorption of BSA on coarse particles, the results were as follows: 100 and 63% for As_4_S_4_ and ZnS samples, respectively.

### 2.5. Mechanisms of Binding

To determine whether albumin could be covalently attached to the surface of ZnS nanoparticles through disulfide bonds, the quantification of S-H bonds have been performed in an experiment using the Ellman reaction. The samples of BSA, ZnS and ZnS–BSA after milling and ZnS–BSA after adsorption were analyzed. It was detected that BSA and ZnS have 0.93 mmol g^−1^ and 1.24 mmol g^−1^ of free mercapto groups, respectively. After milling together, the amount of free mercapto groups decreased to 0.98 mmol g^−1^. On the other hand, in the case of ZnS–BSA after adsorption, the amount of free mercapto groups slightly increased up to value 1.40 mmol g^−1^. The results suggest that there prevails another binding mechanism between BSA and ZnS, not through disulfide bonds (or only partially). The binding of Zn^2+^ on the functional groups of BSA is more probable. As was described above, the FTIR spectra of ZnS conjugated nanoparticles showed some variations in comparison with that of pure BSA. The overlapping stretching vibration of –OH and –NH bonds shifts to higher wavenumbers of about 81 cm^−1^ for ZnS–BSA after milling and 27 cm^−1^ for ZnS–BSA after adsorption. The C-H vibrations at 3060 cm^−1^ were strongly suppressed in a sample of ZnS–BSA after milling and they completely disappeared in the ZnS–BSA after adsorption. In addition, more intense peaks in lower wavenumbers after the interaction of ZnS with BSA (e.g., at 1072 cm^−1^, 986 cm^−1^ and 855 cm^−1^) could correspond to the interaction of Zn and BSA through the hydroxyl and amine groups present in BSA [16]. Moreover, as was mentioned above and also described in our previous paper [14], after the mechanochemical preparation of ZnS nanoparticles, the zeta potential of particles has a positive charge of up to 7.3, which is considerably higher than that found in the literature [41,42,43]. The reason is in the excess of Zn^2+^ ions on the surface of particles, which can interact with the functional groups of BSA. During the mechanochemical synthesis, a lot of defects, cracks, open pores, and intergranular spaces are created at the surface of the samples. The amine and hydroxyl functional groups are originated from lysine [46], tryptophan [74] or other amino acid residues [75], which could be involved in the interaction with zinc from ZnS. Nevertheless, further investigation is required to understand the conjugation mechanism in detail during this process.

### 2.6. Stability Study

#### 2.6.1. Thermal Stability

To evaluate the stability of nanoparticles in a high-temperature environment and analyze the different phases formed by annealing the nanoparticles at higher temperatures, thermal studies were conducted. The obtained thermogravimetry (TG), differential thermogravimetry (DTG), and differential thermal analysis (DTA) graphs are presented in Figure 9. 

In the case of ZnS, the thermogram reveals a weight loss occurring at two distinct maxima: 132 °C and 572 °C, within the temperature range of 20–518 °C and 518–675 °C, respectively (DTG). The first process corresponds to the removal of adsorbed and bound water molecules. The second process involves the exothermic oxidation of ZnS to ZnO, which takes place at approximately 630 °C (DTA). This finding is consistent with the results reported by Dengo et al. [76]. For the ZnS–BSA sample prepared after milling, an additional effect is observed, indicating the decomposition of BSA in the temperature range of 183–575 °C. The TG curve estimates the BSA content in this range to be 18.2 wt.%. However, the precise amount of BSA in the ZnS–BSA sample after adsorption could not be determined due to its limited quantity, which correlates with the FTIR spectroscopy data.

The inclusion of BSA in the samples results in an elevated decomposition temperature of ZnS, following the series: ZnS (519 °C) < ZnS–BSA after milling (579 °C) < ZnS–BSA after adsorption (642 °C). This temperature increase can be attributed to the development of a robust carbon shell during the carbonization of BSA, providing indirect evidence for the presence of BSA in the ZnS–BSA after the adsorption sample. Furthermore, an intriguing phenomenon is observed in the samples containing BSA. Oxygen-deficient ZnO is generated, which undergoes oxidation up to a specific temperature (798 °C for ZnS–BSA after adsorption and 906 °C for ZnS–BSA after milling). During this process, the weight increases due to the addition of oxygen. 

#### 2.6.2. Long-Term Stability

In Figure 10, the particle size distribution and the fluorescence emission spectra measured immediately after the milling (red line) and after one year (blue line) were compared to establish long-term stability. The changes in the shape and/or shift of the curves as a result of sample aging were monitored. As can be seen, after one year of the sample storage in a refrigerator, no dramatic changes were registered. Only, the slight broadening and shift of the Gaussian distribution to larger particles sizes appeared (from d_50_ = 137 nm for sample after milling to d_50_ = 157 nm for sample after one year, Figure 10a) as a consequence of partial dissolution of zinc from ZnS nanoparticles. However, the aggregation of the particles was not observed even after one year. In the case of fluorescence emission spectra (Figure 10b), the higher intensity of the emission band for ZnS–BSA after one year was registered, which confirms the partial dissolution of ZnS nanoparticles. The red shift in the emission band was detected as a consequence of larger particles.

### 2.7. Evaluation of the Cellular Effects of ZnS–BSA in Multiple Myeloma Cells

To evaluate the potential antiproliferative effects of ZnS–BSA in multiple myeloma (MM) cell lines, five MM cell lines (MM.1S, RPMI-S, OPM-1, KMS-11, and OCI-My5 cells) were treated with various concentrations (ranging from 0 to 8 μM) of ZnS–BSA for 24 and 48 h. Subsequently, cell viability has been evaluated using colorimetric MTT assay. The cell viability was uniformly unchanged in all MM cell lines exposed to ZnS–BSA compared to untreated controls at 24 and 48 h (Figure 11). In summary, our data confirmed no significant changes in the antiMM cytotoxic activity of ZnS–BSA. 

To further elucidate the underlying cellular mechanism of ZnS–BSA in MM cells, the dissipation of mitochondrial membrane potential (MMP) in the same MM cell lines after treatment with ZnS–BSA (1–8 μM) was evaluated first. Disruption of MMP is considered one of the earliest hallmarks of apoptosis and the formation of JC-1 dimers is directly proportional to MMP, while their conversion into JC-1 monomers within dying cells is associated with the loss of MMP. We did not observe any notable reduction in MMP in ZnS–BSA-treated MM cells after 48 h, as evidenced by no significant changes in JC-1 monomers compared to controls (Figure 12a). Assessment of apoptosis was further detected by annexin V and PI staining, whereby annexin V staining enables the detection of early apoptosis characterized by the externalization of transmembrane phosphatidylserine. On the other hand, PI staining allows the identification of necrosis, wherein PI is internalized into the nucleus of dead cells. In addition, moderate staining of both fluorochromes indicates the presence of late apoptosis events. Treatment of all MM cell lines with ZnS–BSA (1–8 μM) revealed no significant changes in the induction of apoptosis (both early and late) and necrosis (Figure 12b). Furthermore, the cell cycle profile of ZnS–BSA-treated MM cell lines versus control cells by flow cytometric analysis was evaluated. No significant observations of cell cycle modulation, indicating no changes in the distribution of G1, S and G2/M phases of the cell cycle after ZnS–BSA dose-dependent (1–8 μM) exposure in all tested MM cells at 48 h, were revealed (Figure 12c). Overall, ZnS–BSA did not trigger any significant changes in the cell cycle profile, disruption of MMP and also in the induction of cell death using fluorescence- and colorimetry-based cell assays. However, further investigation is required to ensure the safety and performance of the system for future in-vivo applications.

## 3. Materials and Methods

### 3.1. Synthesis of ZnS Nanoparticles

The stoichiometric mixture of zinc acetate and sodium sulfide were comilled in a planetary mill Pulverisette 6 (Fritsch, Idar-Oberstein, Germany) in order to prepare 3 g of corresponding zinc sulfide according Equation (1). The following milling conditions were used: ball charge 50 balls of 10 mm diameter, ball and vial material tungsten carbide, 10 min milling time in argon atmosphere and rotational speed of the planet carrier 500 min^−1^. After milling, the washing, decantation, filtration and drying were applied to obtain zinc sulfide in solid state. The overall flow chart of the preparation process is shown in [8].
(CH_3_COO)_2_Zn + Na_2_S → ZnS + 2CH_3_COONa(1)

### 3.2. Preparation of ZnS–BSA Nanosuspension

The nanosuspension was prepared in a laboratory circulation mill MiniCer (Netzsch, Selb, Germany). Three grams of ZnS nanocrystalline sample was subjected to a wet milling process in the presence of 300 mL BSA phosphate buffer solution (1 wt.%, pH = 7.4) for 45 min at the milling speed of 3000 rpm. The mill was loaded with yttrium-stabilized ZrO_2_ balls, 0.6 mm in diameter. The resulting nanoparticle suspension was centrifuged at 2000 rpm. Afterwards, the nanosuspension was characterized and stored in a refrigerator (4 °C).

### 3.3. Adsorption Studies

This study investigated the adsorption of bovine serum albumin (BSA, for biochemistry, protease free, Acros Organics, Geel, Belgium) to ZnS nanoparticles in an aqueous buffer solution (pH = 7.4) made with phosphate-buffered saline tablets (Fisher Scientific, Hampton, NH, USA). Kinetic studies were conducted using solutions of BSA with a concentration of 0.5%. The adsorption process was examined at 15 min intervals, covering a time range from 15 to 120 min, utilizing a static mode setup. The experimental parameters included a sample mass (m) of 0.01 g, a volume (V) of 10 mL and a temperature (T) of 25 °C throughout the experiments. Once the desired adsorption time had passed, the samples were filtered, and the resulting filtrates were collected for further analysis. The UV spectra of these filtrates were recorded to gather relevant data and insights. The adsorption kinetics were characterized using two rate laws: the pseudo-first-order model represented by
(2)$$\ln\left(A_{eq}-A_{t}\right)=lnA_{eq}-k_{1}t$$
and the pseudo-second-order model represented by
(3)$$\frac{t}{A_{t}}=\frac{1}{k_{2}A_{eq}^{2}}+\frac{t}{A_{eq}}$$

In these equations, *A_eq_* and *A_t_* stand for the adsorption capacities, which indicate the amount of BSA adsorbed on ZnS at equilibrium and at a specific time at equilibrium and at time *t*; *k*_1_ and *k*_2_ are pseudo-first and pseudo-second order rate constants and *h* = *k*_2_*A_eq_*^2^ is an initial sorption rate assuming pseudo-second kinetic order model.

In order to ascertain the sorption capacity, the adsorption isotherm was investigated using BSA solutions with concentrations ranging from 0.1% to 1% (increasing in 0.1% increments). The contact time between ZnS nanoparticles and BSA solutions was set at 120 min. The Langmuir and Freundlich isotherm models were used to relate the adsorption data. In the Langmuir isotherm, the equation is represented as
(4)$$\frac{C_{eq}}{A_{eq}}=\frac{1}{K_{L}A_{max}}+\frac{C_{eq}}{A_{max}}$$
and on the other hand, the Freundlich isotherm is given by the equation
(5)$$lnA_{eq}=lnK_{F}+\left(\frac{1}{n}\right)lnC_{eq}$$
where *C_eq_* is the equilibrium concentration of BSA in the solution measured in mg L^−1^; *A_eq_*—adsorption capacity at equilibrium, mg g^−1^; *K_L_* is the Langmuir adsorption equilibrium constant; *A_max_*—maximal adsorption capacity for complete monolayer covering of the surface; *K_F_*—Freundlich constant; *n* in the Freundlich equation is an empirical parameter connected to the intensity of adsorption and heterogeneity of the adsorbent. All the experiments were repeated at least three times. The data presented are mean ± standard error.

### 3.4. Characterization Methods

#### 3.4.1. Particle Size Distribution

The particle size distribution was measured by a photon cross-correlation spectroscopy using a Nanophox particle size analyzer (Sympatec, Clausthal-Zellerfeld, Germany). A portion of nanosuspension was diluted with the BSA-containing solution to achieve a suitable concentration for the measurement. This analysis was performed using a dispersant refractive index of 1.33. The measurements were repeated three times for each sample.

#### 3.4.2. Zeta Potential

The zeta potential (ZP) was measured using a Zetasizer Nano ZS (Malvern, Worcestershire, Great Britain). The zetasizer measures the electrophoretic mobility of the particles, which is converted to the ZP using the Smoluchowski equation built into the Malvern zetasizer 7.1 software. The ZP was measured in the original dispersion medium, and the measurements were repeated three times for each sample.

#### 3.4.3. UV-Vis Specroscopy

The absorption spectra were recorded using a UV–Vis spectrophotometer Helios Gamma (Thermo Electron Corporation, Altrincham, Great Britain) in the range of 200–800 nm. The concentration of BSA was determined at λ = 278 nm using a 1-cm lightpath quartz cell. 

The absorption data were analyzed using Benesi–Hildebrand equation [77]
(6)$$\frac{1}{A_{obs}-A_{0}}=\frac{1}{A_{c}-A_{0}}+\frac{1}{K_{app}\left(A_{c}-A_{0}\right)\left[c\right]}$$
where *A_obs_* is the absorbance of BSA with different concentrations of ZnS nanoparticles, *A*_0_ is the absorbance of BSA alone, *A_c_* is the absorbance in the presence of ZnS nanoparticles at 278 nm, *K_app_* is the apparent association constant and *c* is the concentration of ZnS nanoparticles.

#### 3.4.4. Photoluminescence Spectroscopy

The room temperature fluorescence quenching spectra were acquired at the right angle on a photon counting spectrofluorometer PC1 (ISS Inc., Champaign, IL, USA) with an excitation wavelength of 280 nm. A 300 W xenon lamp was used as excitation source. Excitation and emission slit widths were set to 0.5 and 1 mm, respectively. For the measurements, 1 cm path length rectangular quartz cuvette was used. For the measurements of synchronous fluorescence spectra, the intervals between maximum excitation and emission wavelength were set as Δλ = 15 nm and Δλ = 60 nm for tyrosine and tryptophan residues, respectively. 

The quenching mechanism was analyzed by the classical Stern–Volmer equation and the Stern–Volmer constant *K_sv_* was calculated:(7)$$\frac{F_{0}}{F}=1+k_{q}\tau_{0}\left[c\right]=1+K_{SV}\left[c\right]$$
where *F*_0_ is the fluorescence intensity for BSA alone and *F* is the fluorescence intensity in the presence of quencher (ZnS nanoparticles), *k_q_* is bimolecular quenching rate constant, *K_sv_* is the Stern–Volmer constant, *τ_0_* is the average lifetime of the BSA and *c* is the concentration of ZnS nanoparticles.

The bimolecular quenching rate constant *k_q_* was calculated using the formula:(8)$$k_{q}=\frac{K_{S}}{\tau_{0}}$$
where *τ*_0_ is lifetime of BSA (4.63 × 10^−8^ s).

The number of ZnS molecules (*m*) that interact simultaneously with each site of BSA and the apparent binding constant (*K_m_*) were obtained using the equation for the determination of stoichiometry of the binding process [29]
(9)$$log\left[\frac{F_{0}-F}{F}\right]=logK_{m}+mlog\left[c\right]$$

The accessibility of BSA to ZnS nanoparticles was calculated by the modified Stern–Volmer equation [78]
(10)$$\frac{F_{0}}{F_{0}-F}=\frac{1}{f_{a}}+\frac{1}{K_{a}f_{a}\left[c\right]}$$
where *f_a_* is the fraction of fluorophore (BSA) accessible to the quencher (ZnS nanoparticles), and *K_a_* is the effective quenching constant for the accessible fluorophores.

#### 3.4.5. Circular Dichroisms Spectroscopy

A Jasco J-810 spectropolarimeter (Japan) was used for the measurements of circular dichroism (CD) spectra. The room temperature spectra were recorded in the range of 195–250 nm at a scan speed of 50 nm min^−1^. One mm path length quartz cuvette was used for the measurements. Each spectrum was an average of 3 scans after baseline substraction using a PBS buffer (pH 7.4) as the reference baseline, with a data pitch of 1 nm, an integration time of 2 s and a bandwidth of 2 nm.

#### 3.4.6. Fourier-Transform Infrared Spectroscopy 

A Tensor 27 (Bruker, Mannheime, Germany) was used for the measurements of Fourier transform infrared (FTIR) spectra. The spectra were recorded in the range of 4000–400 cm^−1^ using a KBr pellet method. KBr was dried 24 h at 100 °C before the analysis.

#### 3.4.7. Thermogravimetry

A derivatograph Q–1500 D apparatus (Paulik, Paulik and Erdey, MOM, Budapest, Hungary) was used for the measurements of the thermograms (thermogravimetry, TG at the average weight errors ±0.1 mg; differential TG, DTG; and differential thermal analysis, DTA). The data were recorded using upon heating of samples (~0.1 g) in air at a heating rate of 10 °C/min from room temperature to 1000 °C.

#### 3.4.8. Determination of –SH Groups

To assess the accessibility of the -SH groups, the Ellmann reaction was employed [79]. The number of sulfhydryl groups was determined using a reagent known as Ellman’s reagent, specifically 5,5′-dithio-bis(2-nitrobenzoic) acid (99%, Acros organics, Geel, Belgium). The reaction involves a thiol–disulfide exchange with the surface -SH groups, which results in the formation of the *p*-nitrobenzoate anion in the solution, exhibiting a strong absorption peak at 412 nm. In the procedure, approximately 50 mg of the sample was mixed with 50 mL of Ellman’s reagent (100 mg of Ellman’s reagent was dissolved in 50 mL of MeOH). To this mixture, 500 μL of triethylamine (TEA; 99%, Acros Organics, Geel, Belgium) was added, and the flask was shaken for 30 min. The solid components were then filtered out, and the filtrate was further diluted with MeOH for UV analysis.

### 3.5. Cell Cultures Studies

A collection of multiple myeloma (MM) cell lines, including MM.1S, RPMI-S (also known as RPMI 8226-S), OPM-1, KMS-11 and OCI-My5 cells, was obtained from the American Type Culture Collection (Manassas, VA, USA). MM cell lines were cultured in RPMI 1640 medium (Cellgro, Mediatech, VA, USA) supplemented with 10% heat-inactivated fetal bovine serum (FBS; Harlan, Indianapolis, IN, USA), 100 μg mL^−1^ penicillin, 100 μg mL^−1^ streptomycin and 2 mM L-glutamine (GIBCO, Grand Island, NY, USA). The cultures were maintained at 37 °C in a 5% CO_2_ atmosphere.

#### 3.5.1. Spectrophotometry-Based Colorimetric MTT Cell Assay

MM cells were seeded at 1 × 10^4^ cells/well (in 100 μL of medium) in optical 96-well plates and at 24 h after plating treated with ZnS–BSA at various concentrations (0.125, 0.25, 0.5, 1, 2, 4, and 8 μM in 100 μL of medium) for 24 and 48 h. The inhibitory effect of ZnS–BSA on survival of MM cells was evaluated by adding 50 μL (1 mg mL^−1^) of 3-[4,5-dimethylthiazol-2-yl]-2,5-diphenyltetrazolium bromide assay (MTT; Sigma-Aldrich, St Louis, MO, USA) per well. Subsequently, formazan crystals produced after 4 h were solubilized using 150 μL of dimethyl sulfoxide (DMSO). The resulting solution was subjected to spectrophotometric analysis, measuring absorbance at 540 and 690 nm in a microplate reader (Dynatech Lab Inc., Chantilly, VA, USA). 

#### 3.5.2. Flow-Cytometry Based Fluorescence Cell Assays 

The cellular mechanism of ZnS–BSA treatment in MM cells were evaluated by fluorescence assays: annexin V-fluorescein isothiocyanate (Annexin V-FITC) assay for quantification of apoptosis; assessment of mitochondrial membrane potential using JC-1 fluorescent probe; and DNA content analysis of nuclei labelled with propidium iodide (PI) for evaluation of cell cycle changes. The mitochondrial membrane potential of ZnS–BSA-treated (1, 2, 4 and 8 μM) MM cells versus control cells was assessed at 48 h using JC-1 fluorescent probe. Cells (3 × 10^5^) were incubated with 4 μM JC-1 (Molecular Probes, Eugene, OR, USA) in 200 μL of PBS/0.2% BSA for 30 min in the dark at 37 °C. The induction of apoptosis in ZnS–BSA-treated (1, 2, 4, and 8 μM) MM cells versus control cells was evaluated at 48 h using Annexin V-FITC and propidium iodide (PI) apoptosis assay kit. Cells (3 × 10^5^) were resuspended in 100 μL of 1× binding buffer and stained with 5 μL of Annexin V-FITC and 5 μL of PI for 30 min at room temperature. Changes in the cell cycle status of ZnS–BSA-treated (1, 2, 4, and 8 μM) MM cells compared to control cells were assessed at 48 h by measuring the DNA content of nuclei labeled with PI dye. Cells (3 × 10^5^) were incubated in 0.05% Triton X-100 and 15 μL of RNAse A (10 mg mL^−1^) for 20 min at 37 °C and then, cooled on ice for 10 min before PI (50 μg mL^−1^) was added. Flow cytometry measurements were performed using a FACS Canto II flow cytometer equipped with a 488 nm excitation laser. The fluorescence emission was detected using specific bandpass filters (530 nm, 585 nm, 670 nm and 780 nm) and corresponding photomultipliers (FL1-FL4) for different fluorochromes. All fluorochromes were excited with the 488 nm laser, and the emitted fluorescence signals were collected by the respective photomultipliers as follows: Annexin V-FITC and PI (FL1, FL2), JC-1 (FL1, FL2, and FL2/FL1 ratio) and cell cycle analysis (log FL3 for sub G1, linear FL2 for DNA cell cycle histogram, FL3 peak versus integral for doublet discrimination). The forward/side scatter characteristics were utilized to exclude cell debris from the analysis. Approximately, 1 × 10^4^ cells were acquired for each analysis. Data analysis was performed using De Novo FCS Express software (De Novo software, Los Angeles, CA, USA), and cell cycle calculations were determined using MultiCycle AV DNA analysis (Phoenix Flow Systems, San Diego, CA, USA).

#### 3.5.3. Statistical Analysis

The statistical significance of differences between ZnS–BSA-treated and control samples in the in vitro studies was assessed using one-way ANOVA. The significance levels were denoted as *p* < 0.05. The data are presented as mean ± standard deviation.

## 4. Conclusions

In this study, a highly stable nanosuspension of zinc sulfide bioconjugated with albumin was prepared with mechanochemical approaches. The sorption kinetics were well fitted by the pseudo first-order model, and the adsorption parameters were better described by the Freundlich isotherm model. The sorption capacity was determined to be 120 mg g^−1^. The interactions between ZnS and albumin were studied using fluorescence spectroscopy and infrared spectroscopy. The fluorescence spectra analysis revealed both dynamic and static quenching, with Stern–Volmer plots showing a nonlinear trend with an upward curvature concave towards the y-axis. The Stern–Volmer constant was calculated from the initial linear part of the regression curve appertaining lower quencher concentrations to be 2 × 10^4^ M^−1^. Other parameters describing the interactions of BSA with mechanosynthesized ZnS nanoparticles were calculated including the binding constant, stoichiometry of binding process and accessibility of the fluorophore to quencher. The analysis of the synchronous fluorescence spectra suggests that the tryptophan can be closer to the binding site than tyrosine residue. The changes in the secondary and tertiary structure of albumin were followed by the infrared spectroscopy, which disclosed only slight changes. The 6% decrease of α-helix content was calculated after the deconvolution of the Amide I band, which is the most sensitive probe for detecting changes in the protein secondary structures. Furthermore, the fluorescence techniques were also used for the evaluation of the cellular mechanism in multiple myeloma cells treated with ZnS–BSA nanosuspension (quantification of apoptosis, assessment of mitochondrial membrane potential and evaluation of cell cycle changes). The preliminary results confirmed the nontoxic nature of ZnS nanoparticles, making them potentially applicable in drug delivery systems. However, further investigation is required to ensure the safety and performance of the system for in vivo applications. Overall, the fluorescence techniques proved to be very useful tools for the study of the *nano–bio* interactions, i.e., interactions between (in)organic *nano*particles and *bio*materials.

## Figures and Tables

**Figure 1 pharmaceuticals-16-01219-f001:**
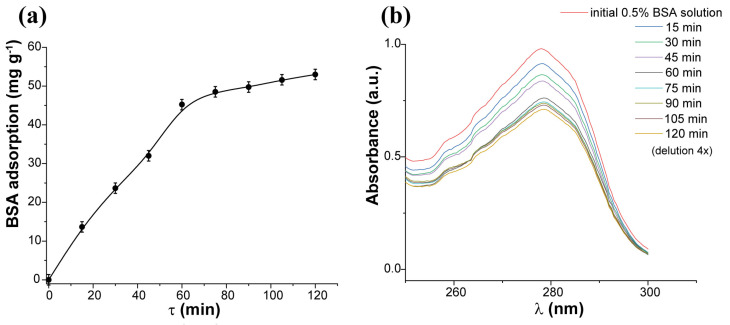
(**a**) The BSA adsorption kinetic curve and (**b**) UV–Vis spectra at each kinetic point.

**Figure 2 pharmaceuticals-16-01219-f002:**
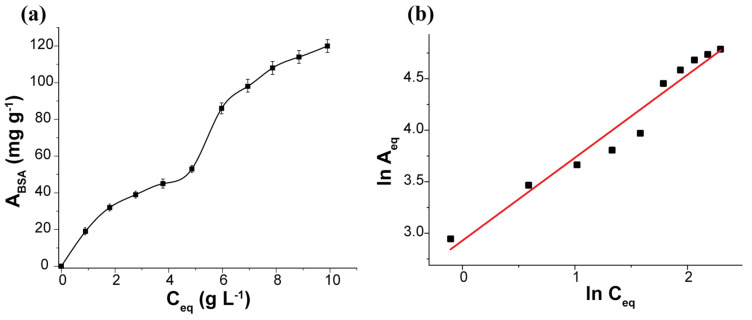
(**a**) The isotherm of BSA adsorption by ZnS nanoparticles and (**b**) its linearized graph in the coordinates of the Freundlich (lnA_eq_ vs. lnC_eq_) isotherm equation.

**Figure 3 pharmaceuticals-16-01219-f003:**
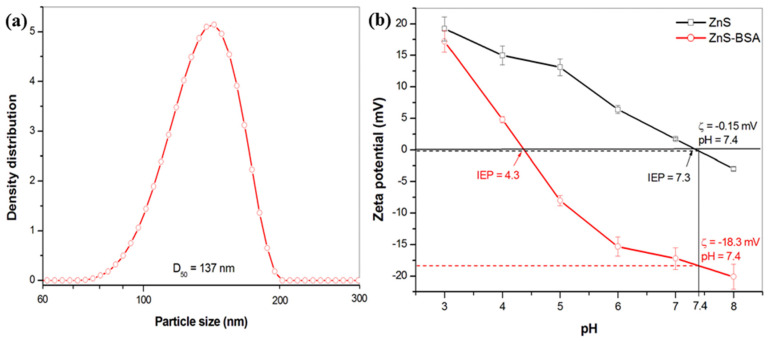
(**a**) Particle size distribution, and (**b**) dependence of zeta potential vs. pH of ZnS and ZnS–BSA samples.

**Figure 4 pharmaceuticals-16-01219-f004:**
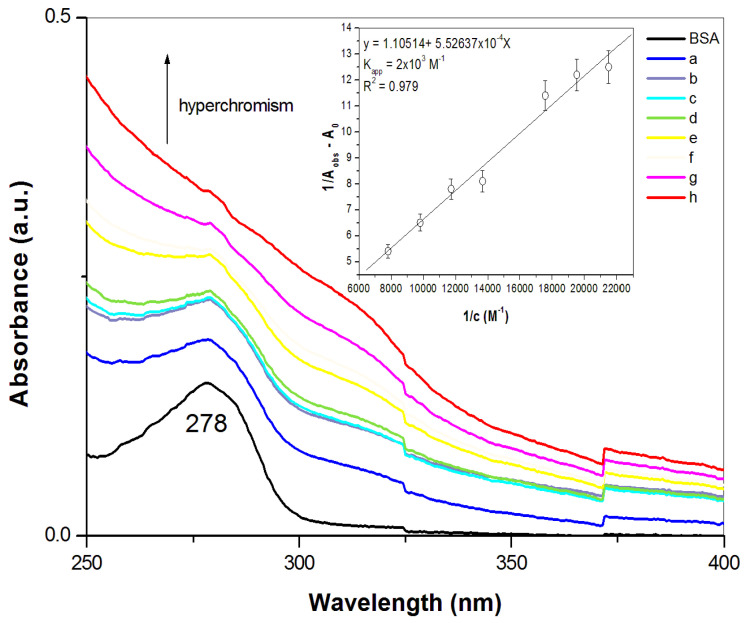
UV–Vis absorption spectra of BSA (2.5 × 10^−5^ M) and BSA with different concentrations of ZnS nanoparticles (a = 4.3 × 10^−5^ M, b = 4.7 × 10^−5^ M, c = 5.1 × 10^−5^ M, d = 5.7 × 10^−5^ M, e = 7.3 × 10^−5^ M, f = 8.5 × 10^−5^ M, g = 1.0 × 10^−4^ M, h = 1.3 × 10^−4^); inset: linear dependence of 1/(A_obs_ − A_0_) vs. 1/[c]. All measurements were performed in triplicate.

**Figure 5 pharmaceuticals-16-01219-f005:**
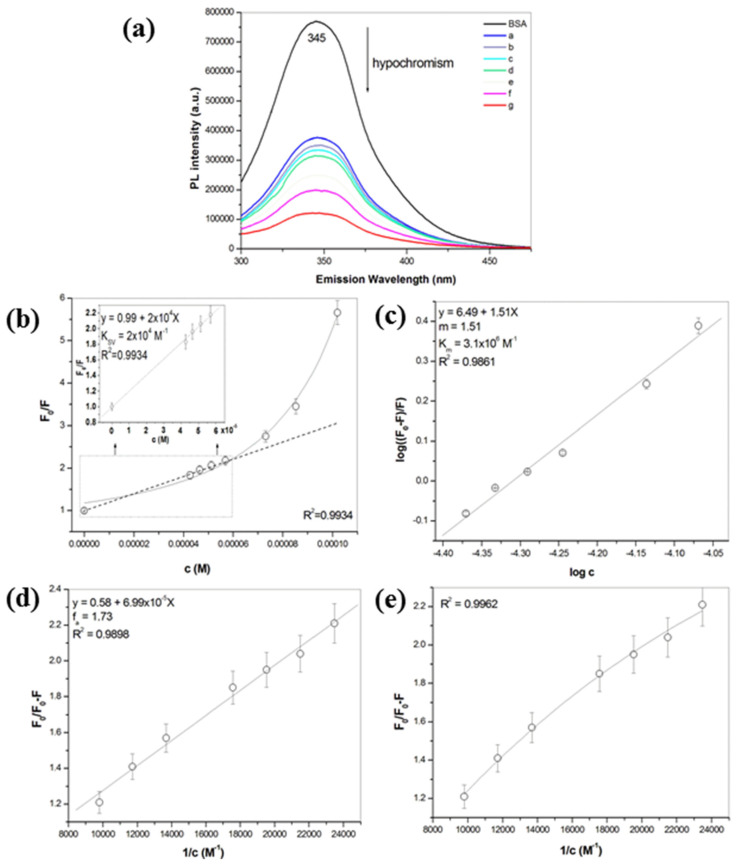
(**a**) Fluorescence emission spectra of BSA (2.5 × 10^−5^ M) and BSA with various concentrations of ZnS nanoparticles (a = 4.3 × 10^−5^ M, b = 4.7 × 10^−5^ M, c = 5.1 × 10^−5^ M, d = 7.3 × 10^−5^ M, e = 8.5 × 10^−5^ M, f = 1.0 × 10^−4^ M, g = 1.3 × 10^−4^ M), (**b**) Stern–Volmer plot F_0_/F vs. [c], (**c**) the plot of log[F_0_ − F/F] vs. log[c], (**d**) linear, and (**e**) first order dependence of F_0_/F_0_ − F vs. 1/[c]. All measurements were performed in triplicate.

**Figure 6 pharmaceuticals-16-01219-f006:**
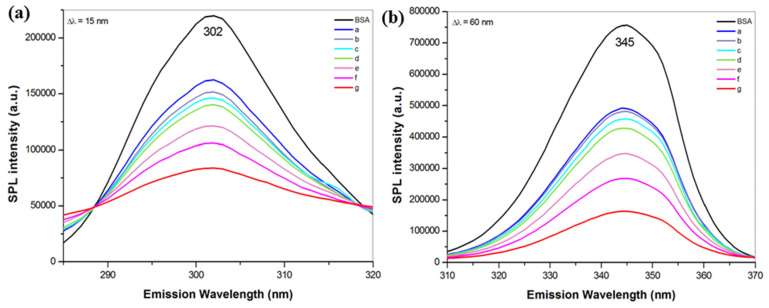
Synchronous fluorescence spectra of BSA (2.5 × 10^−5^ M) and BSA with various concentrations of ZnS nanoparticles (a = 4.3 × 10^−5^ M, b = 4.7 × 10^−5^ M, c = 5.1 × 10^−5^ M, d = 7.3 × 10^−5^ M, e = 8.5 × 10^−5^ M, f = 1.0 × 10^−4^ M, g = 1.3 × 10^−4^ M) at (**a**) Δλ = 15 nm (tyrosine) and (**b**) Δλ = 60 nm (tryptophan). All measurements were performed in triplicate.

**Figure 7 pharmaceuticals-16-01219-f007:**
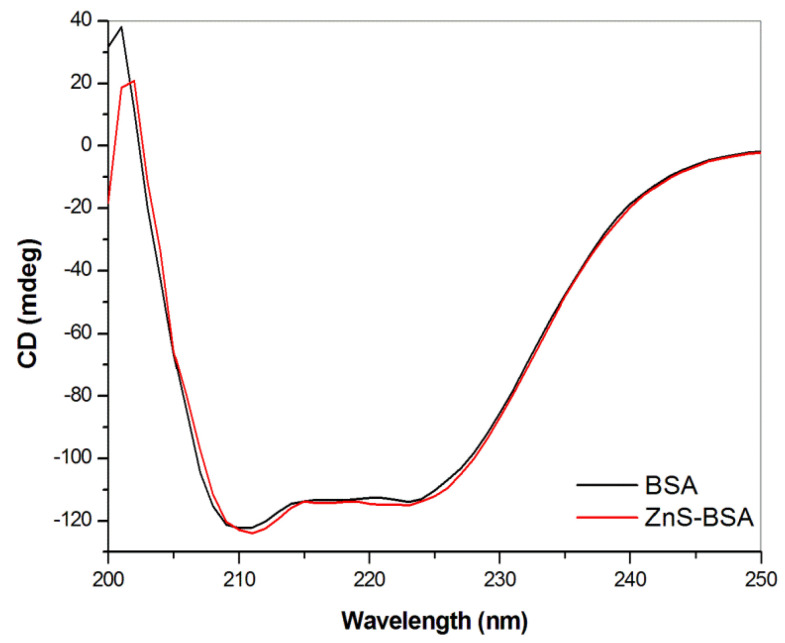
Circular dichroisms spectra of BSA (10 μmol) for BSA and ZnS–BSA. All measurements were performed in triplicate.

**Figure 8 pharmaceuticals-16-01219-f008:**
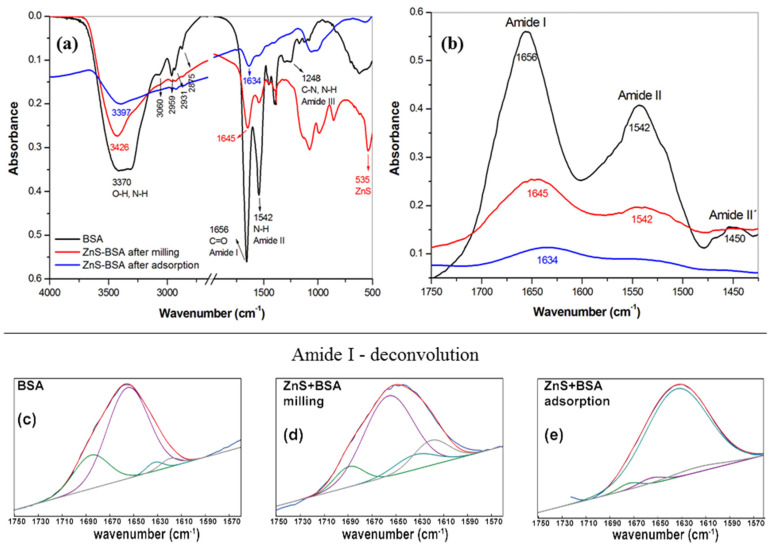
(**a**) FTIR spectra (500–4000 cm^−1^), (**b**) amide I region (1400–1750 cm^−1^), (**c**–**e**) curve fitting in the amide I region with secondary determination of BSA, ZnS–BSA after milling and after adsorption, respectively. All measurements were performed in triplicate.

**Figure 9 pharmaceuticals-16-01219-f009:**
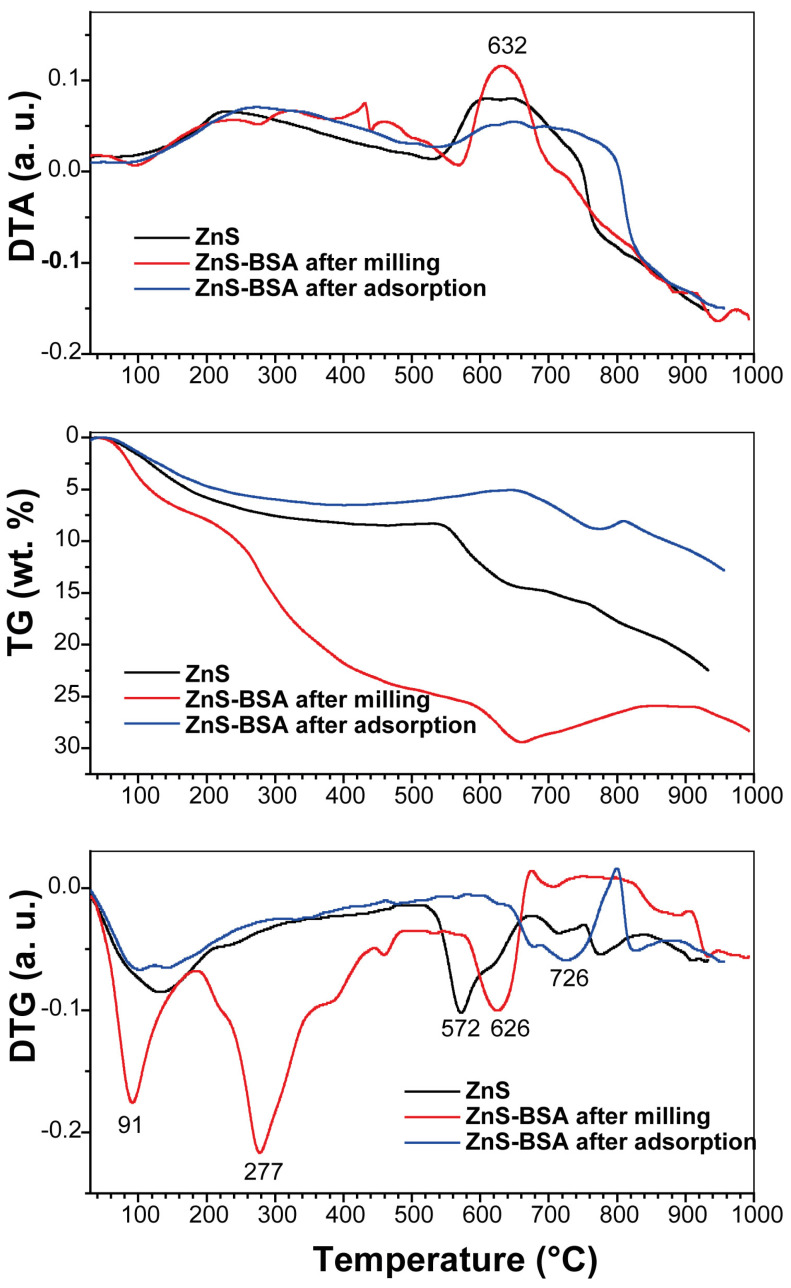
DTA, TG and DTG curves of ZnS and ZnS–BSA after milling and after adsorption samples.

**Figure 10 pharmaceuticals-16-01219-f010:**
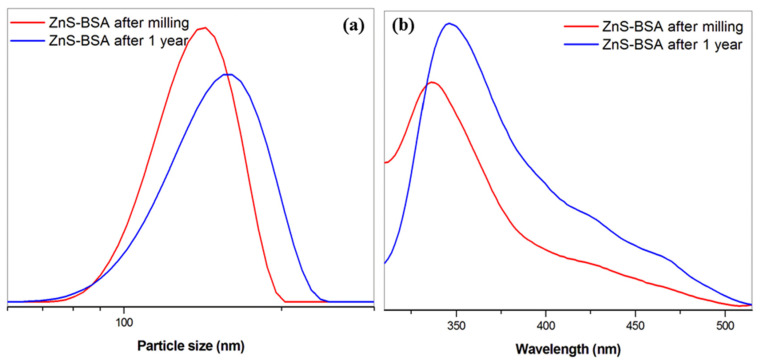
Long-term stability studies of ZnS–BSA measured immediately after milling and after 1 year, (**a**) particle size distribution, (**b**) fluorescence emission spectra. All measurements were performed in triplicate.

**Figure 11 pharmaceuticals-16-01219-f011:**
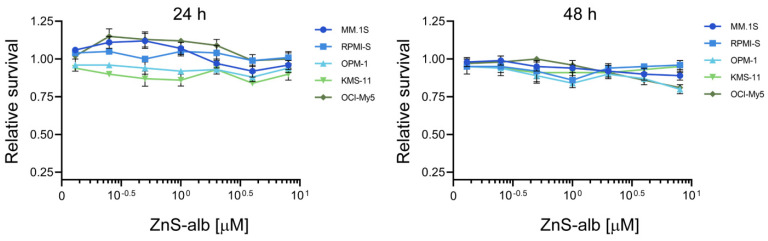
The cytotoxic effects of ZnS–BSA on MM cell lines. MM cell lines (MM.1S, RPMI-S, OPM-1, KMS-11 and OCI-My5 cells) were exposed to ZnS–BSA at various concentrations (0.125, 0.25, 0.5, 1, 2, 4, and 8 μM) for 24 and 48 h. The cell survival was assessed by MTT assay. Each treatment with a specific concentration of ZnS–BSA was performed in triplicate. The presented data represent the mean ± standard deviation of cell survival/viability relative to the untreated controls.

**Figure 12 pharmaceuticals-16-01219-f012:**
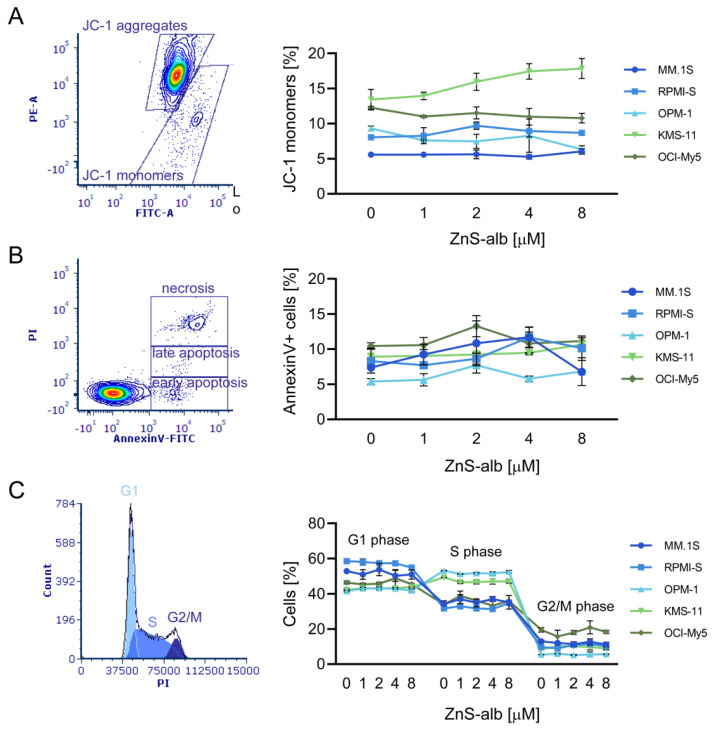
Flow cytometry-based fluorescence analysis of disruption of mitochondrial membrane potential, phosphatidylserine externalization and cell cycle analysis. MM.1S, RPMI-S, OPM-1, KMS-11 and OCI-My5 cells were cultured with ZnS–BSA at concentrations of 1, 2, 4 and 8 μM for 48 h. (**A**) In dot plot, JC-1 dye forms aggregates in normally polarized mitochondria, emitting an orange fluorescence (JC-1 aggregates gate). In cells with depolarized mitochondrial membranes, JC-1 exists in monomeric form and emits green fluorescence (JC-1 monomers gate) in ZnS–BSA-treated OPM-1 cells at 8 μM for 48 h. In the graph, production of JC-1 monomers representing disruption of mitochondrial membrane potential in ZnS–BSA-treated MM cell lines compared to control was quantified by staining with the fluorescent JC1 dye. (**B**) In dot plot, early apoptotic (Annexin V-FITC+/PI-; early apoptosis gate), late apoptotic (Annexin V+/PI+/−; late apoptosis gate) and necrotic (Annexin V+/PI+; necrosis gate) cells in ZnS–BSA-treated OPM-1 cells at 8 μM for 48 h were determined by Annexin V-FITC and PI assay. In the graph, percentage of annexin V+ cells showing induction of apoptosis and necrosis in ZnS–BSA-treated MM cell lines compared to control was evaluated. (**C**) Cell cycle analysis of ZnS–BSA-treated OPM-1 cells at 8 μM for 48 h with the distribution of cells in G1, S and G2/M phase is shown in a representative histogram. The graph shows the distribution of cells in G_1_, S and G2/M phase in MM cells treated with different concentrations of ZnS–BSA compared to untreated controls. The flow cytometry-based fluorescence cell analyses were analyzed using a FACS Canto II flow cytometer and further processed with De Novo FCS Express software (https://denovosoftware.com/). Data are derived from three independent experiments and presented as means ± standard deviation.

**Table 1 pharmaceuticals-16-01219-t001:** The parameters of BSA adsorption by ZnS sample.

**The BSA Kinetic Sorption Parameters**						
**Pseudo-First Order**			**Pseudo-Second Order**			
** *A_eq_* ** **mg g^−1^**	** *k* ** ** _1_ ** **min^−1^**	** *R* ** ** ^2^ **	** *A_eq_* ** **mg g^−1^**	** *k* ** ** _2_ ** **mg^−1^ min^−1^**	** *h* ** **mg min^−1^ g^−1^**	** *R* ** ** ^2^ **
69.7	0.0353	0.968	93.8	0.0001	1.143	0.920
**The BSA adsorption parameters estimated by Langmuir and Freundlich isotherm models**						
**Sorption capacity** **mg g^−1^**	**Langmuir equation**			**Freundlich equation**		
	** *A_max_* ** **mg g^−1^**	** *K_L_* ** **L mg^−1^**	** *R* ** ** ^2^ **	** *K_F_* ** **mg g^−1^**	** *n* **	** *R* ** ** ^2^ **
120.0	408.2	0.041	0.244	18.69	1.24	0.956

**Table 2 pharmaceuticals-16-01219-t002:** Fluorescence quenching constants (calculated from modified Stern–Volmer plot analysis) for the interaction of BSA with ZnS nanoparticles.

*K_app_* (×10^3^ M^−1^)	*K_SV_* (×10^4^ M^−1^)	*k_q_* (×10^11^ M^−1^s^−1^)	*m*	*K_m_* (×10^6^ M^−1^)	*f_a_*	*K_a_* (×10^3^ M^−1^)
2	2	4.3	1.51	3.1	1.73	8.3

**Table 3 pharmaceuticals-16-01219-t003:** BSA secondary structure determination.

	BSA		ZnS–BSA after Milling		ZnS–BSA after Adsorption	
	Vibration (cm^−1^)	Content (%)	Vibration (cm^−1^)	Content (%)	Vibration (cm^−1^)	Content (%)
antiparallel β-sheet	1686	22	1689	10	1680	2
α-helix	1655	72	1656	66	1660	3
random coil	1633	3	1634	10	1637	91
β-sheet	1620	3	1620	14	1618	4

## Data Availability

The data presented in this study are available on request from the authors.

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
