# Peer review of "Investigation of the Interaction between Mechanosynthesized ZnS Nanoparticles and Albumin Using Fluorescence Spectroscopy"

_pharmaceuticals, 2023, doi:10.3390/ph16091219_

Round 1

Reviewer 1 Report

The article reported a mechanical stirring method to synthesize bovine serum albumin conjugated ZnS. The as-synthesized particles were thoroughly characterized by fluorescence techniques to study the interaction between albumin and ZnS. Specifically, the quenching mechanism, binding constant and stoichiometry of protein complex was calculated and analyzed based on the Stern-Volmer equation. Thermal stability was evaluated by thermogravimetry (TG), differential thermogravimetry (DTG), and differential thermal analysis (DTA). In addition, the cytotoxicity study showed no toxicity of the ZnS-BSA compound on different myeloma cell lines. While the authors demonstrate many results to support the conclusion, some questions need to be addressed in this article (see comments below).

1.     In section 3.1, a minireview was presented to show the properties of mechanochemically synthesized ZnS from the citations. Can the authors show original data, such as HRTEM and XRD? Whether these original data match the published results?

2.     What are the UV-vis absorption spectra of BSA, ZnS nanoparticles, and ZnS-BSA bioconjugate?

3.     What’s the milling temperature? Did milling cause an increase in temperature? Whether the milling process affects the integrity of BSA?

4.     On Page 20 lines 717-709, please clarify the exact size of BSA-ZnS before and after 1-year storage.

5.     In Figure 10b and the corresponding discussion on lines 710-712, the authors claim that the peak shift was caused by the dissolution of ZnS. Does any paper support the conclusion? What’s the effect of the dissociation of BSA? Please clarify. 

Minor editing of English language required

Reviewer 2 Report

The overall structure of the article"Investigation of the interaction between mechanosynthesized ZnS nanoparticles and albumin using a fluorescence spectrum" is clear, which significantly contributes to the understanding of the interaction mechanism between albumin and mechanosynthesized ZnS nanoparticles. However, some areas require refinement. For instance, certain figures exhibit blurred horizontal and vertical coordinates, which could be improved for better clarity and presentation. Additionally, some units have errors, such as the conclusion stating 2 x 104 M-1. It is suggested to carefully review and correct these inaccuracies.

Furthermore, the data related to the correlation coefficient of the design formula needs to be repeated, and error bars should be included to enhance the statistical reliability and precision of the results.

Overall, the article provides valuable insights into the interaction between mechanosynthesized ZnS nanoparticles and albumin, but addressing these small details would further enhance its quality and impact.

Overall, the passage demonstrates a reasonably good quality of the English language. The information is conveyed clearly, and the key findings are presented concisely. However, some minor grammatical and stylistic improvements can be made for further clarity and coherence:

For instance, the following change may be considered for the abstract.

"The quenching mechanism was investigated, describing both the static and dynamic interactions." -> Consider rephrasing for clarity: "The quenching mechanism, describing both static and dynamic interactions, was investigated."

"Various parameters describing the interaction of albumin with mechanosynthesized ZnS nanoparticles were calculated (e.g. quenching rate constant, binding constant, stoichiometry of the binding process, and accessibility of fluorophore to the quencher)." -> Consider using a colon to introduce the list of parameters for improved readability: "Various parameters were calculated, including quenching rate constant, binding constant, stoichiometry of the binding process, and accessibility of fluorophore to the quencher."

"The synchronous fluorescence spectra analysis indicated that the fluorescence from tryptophan is more intense and gets more efficiently quenched than those from tyrosine residues in the presence of ZnS..." -> Rephrase for clarity and consistency: "The synchronous fluorescence spectra analysis indicated that the fluorescence from tryptophan is more intense and is more efficiently quenched than that from tyrosine residues in the presence of ZnS..."

"The cellular mechanism in multiple myeloma cells treated with the nanosuspension was evaluated by fluorescence assays for quantification of apoptosis, assessment of mitochondrial membrane potential, and evaluation of cell cycle changes confirming the non-toxic nature of ZnS nanoparticles potentially applicable in the drug delivery system." -> Divide this sentence into two for better readability and clarity. 

"Additionally, the slight changes in the secondary structure of albumin accompained with a decrease in α-helix content were investigated using the FTIR method after analyzing the deconvoluted Amide I band spectra of ZnS nanoparticles conjugated with albumin." -> Consider rephrasing: "Additionally, slight changes in the secondary structure of albumin, accompanied by a decrease in α-helix content, were investigated using the FTIR method after analyzing the deconvoluted Amide I band spectra of ZnS nanoparticles conjugated with albumin."

"The nanosuspension demonstrated stability for over one year." -> This sentence seems out of context. Consider repositioning it to a more appropriate location, or provide more information regarding the relevance of this statement.

Reviewer 3 Report

The authors presented the paper "Investigation of the interaction between mechanosynthesized ZnS nanoparticles and albumin using a fluorescence spectroscopy"

1) The reference list should be improved. I see that many works may be changed for a fresh one, showing the progress of the area in the Introduction section. I highly recommend not to use the references older 10 years for all sections, except for historically important works. Many 2020-2023 papers are greatly suitable for your work. 

2) The explanation of aim of the work and maybe the whole aim should be impoved. In the Introduction of the paper I see ref. 4 which shows the data about some ZnS quantum dots for chemo/biosensing and bioimaging. However, most of the data from this review is the detection of various analytes in water solution and no more. Moreove, ZnS in ref. 8 presents high sized clusters but not small nanoparticles or quantum dots. It is not the same material at all.

3) Moreover, I have found some more papers about albumin coated ZnS quantum dots, which describes various properties. However, the authors don't compare their data with the references, which can be found in the Introduction, and don't desribe such work results. The authors' results should be compared with a literature data. In this way, the novelty of the work should be clearly mentioned in the Results and Discussion, Conclusion section and Abstract. 

4) The Materials and Methods section should be divided into several subsection according to the data. For example, Materials, Synthesis, Various mentods, cell studies, etc. Experimental data about ZnS synthesis in ref. 8 is poor and should be presented in the present work in an extensive way one more time.

5) Section 3.1. Ref. 41-43 present mostly not simple ZnS nanoparticles, and the synsthesis of ZnS is not mentioned in an extensive way. Please, present all the spectra in SI to be sure the material propeties for the readers. 

6) Please, present in SI the figures for Table 1 with approximation curves. How you explaine the Pseudo-first order and Pseudo-second order mechanism? Aeq is much different. Please, add some explanation, disscussion of such results with a comparison with a literature.

Fig 2a. What happens at 5 g/L Ceq?

7) Fig3a. Please, mention the DLS mode size, volume or intensity. Is it ZnS-BSA nanoparticles (mention in the caption)? What was the initial size of ZnS?

8) Fig 7 CD spectrum. I see you have a problem in a 200-205 region which is highly important for alpha helix calculation. Please, present HT channel. There can be an off-scale in the spectrum. Moreover, albumin usually have no more than 63-67 % of alpha helix (Table 3, 72 %). Please, present in SI CD cpestra with your approximation for all spectra. Moreover, I see β-sheet content ~ 3% which is good. However what does mean antiparallel β-sheet ~ 22% in such helical protein (may be it is random coil or turns?). What is the differrenses beetween the calculation of β-sheet and antiparallel β-sheet calculation. Present theoretical CD curves for such species or check/ensure your calculations.

9) Section 3.6.1 Please, add some discussion why the authors requires such experiment for their in vivo application, which will works at room temperature.

Section 3.6.2. Please, clearly mention the storage conditions (in solid, in water, pH, temperature, etc.). It is an essential thing for protein based nanoparticles. The size may be the same but the protein may be spoiled, bad smelled, or eaten by bacteria (expecially in an aqueous conditions). Have you got any biological data of storaged nanoparticles?

10) Limitations of the work should be clearly mentioned in Conclusion section. At the moment, I see synthesis work. Authors should clearly mention nanoparticles application elsewhere (ex. cancer treatment, etc.) and study the appropriate and required properties. However, many data are required to ensure the safety and performance of the system for in vivo applications.  Such simple cytotoxicity test may not be relevant, which was shown by a number of works.

Moderate editing of English language required

Round 2

Reviewer 3 Report

Thank you for the revised version.

Minor editing of English language required